# Development and validation of audio-based guided imagery and progressive muscle relaxation tools for functional bloating

Vincent Tee[1], Garry Kuan[2,3], Yee Cheng Kueh[4], Nurzulaikha Abdullah[4], Kamal Sabran[5], Nashrulhaq Tagiling[6], Nur-Fazimah Sahran[1], Tengku Ahmad Iskandar Tengku Alang[1], Yeong Yeh Lee[1,7]*

1 Department of Medicine, School of Medical Sciences, Universiti Sains Malaysia, Kota Bharu, Kelantan, Malaysia, 2 Exercise and Sport Science Programme, School of Health Sciences, Universiti Sains Malaysia, Kota Bharu, Kelantan, Malaysia, 3 Department of Life Sciences, Brunel University, London, United Kingdom, 4 Biostatistics and Research Methodology Unit, School of Medical Sciences, Universiti Sains Malaysia, Kota Bharu, Kelantan, Malaysia, 5 School of Arts, Universiti Sains Malaysia, Georgetown, Pulau Pinang, Malaysia, 6 Department of Nuclear Medicine, Radiotherapy and Oncology, School of Medical Sciences, Universiti Sains Malaysia, Kota Bharu, Kelantan, Malaysia, 7 GI Function & Motility Unit, Hospital USM, Universiti Sains Malaysia, Kubang Kerian, Kelantan, Malaysia

* justnleeyy@gmail.com, yylee@usm.my

**Data Availability Statement:** All relevant data are within the paper and its Supporting Information file. The full datasets are not publicly available due to de-identified data sharing restrictions by the Human Research Ethics Committee of Universiti

## Abstract

Mind-body techniques, including Guided Imagery (GI) or Progressive Muscle Relaxation (PMR), may effectively manage bloating. The current study aimed to develop and validate (psychometric and psychological responses) audio-based GI and PMR techniques for bloating. Audio scripts were first developed from literature reviews and in-depth interviews of participants with bloating diagnosed based on the Rome IV criteria. Scripts were validated using psychometric (content & face validity index) and physiological approaches (brain event-related potentials & heart rate variability). 45/63 participants completed the in-depth interview, and 'balloon' emerged as the synonymous imagery description for bloating, of which inflation correlated with a painful sensation. The final tools consisted of narrated audio scripts in the background of a validated choice of music. Overall, the content and face validity index for PMR and GI ranged from 0.92 to 1.00. For ERP and HRV, 17/20 participants were analyzed. For ERP, there was a significant difference between GI and PMR for alpha waves (p = 0.029), delta waves (p = 0.029), and between PMR and control for delta waves (p = 0.014). For HRV, GI and PMR exhibited similar autonomic responses over controls (overall p<0.05). The newly developed GI and PMR audio-based tools have been validated using psychometric and physiological approaches.

## Introduction

Abdominal bloating is a common problem faced by 76% of patients with disorders of gut-brain interactions (DGBIs) [1–3]. Bloating is commonly associated with a poor quality of life and well-being, unproductiveness, a greater need to seek healthcare services, and psychological

Sains Malaysia. The reason is that the data contain sensitive information (hospital registration number) and are easily identifiable as they come from one single hospital. Data are, however, available from the corresponding author upon reasonable request and with permission of the Human Research Ethics Committee of Universiti Sains Malaysia. The committee can be contacted at the following website: http://www.jepem.kk.usm.my/.

**Funding:** V.T. is funded by the Malaysian Society of Gastroenterology & Hepatology (MSGH) Research Award 2021. Y.C.K is funded by the Research University Individual Grant from Universiti Sains Malaysia (1001.PPSP.8012250). Y.Y.L. is funded by the Morinaga-USM Gut Health & Microbiota Project Grant, 304/PPSP/6150155/M145. URL of MSGH: https://www.msgh.org.my/. URL of Universiti Sains Malaysia https://www.usm.my/index.php/. The funders had no role in study design, data collection and analysis, decision to publish, or preparation of the manuscript.

**Competing interests:** The authors have declared that no competing interests exist.

disturbance [4–9]. A recent survey found that DGBIs had affected 40% of the global population–highlighting their impact and burden on the healthcare system [10].

Although no treatment is universally accepted for bloating, several interventions have been proposed based on known pathophysiology [11]. One such intervention is based on the biopsychosocial model and multidimensional approach [12]. Due to gut-brain axis dysfunction in bloating, patients have been treated successfully with behavioral and psychological therapies [13–15]. Experts have recommended mind-body techniques such as hypnosis, cognitive-behavioral therapy (CBT), and mindful breathing [16].

Guided Imagery (GI) is an individualized hypnotic-like mind-body technique commonly employed in sports psychology [17, 18]. The contents of GI may be visual, sound, or sensation and are guided using senses to create a reliable, consistent, and specific imagery in the minds [19–21]. Imagery may be used to transform negative sensations like pain and tension into positive feelings like comfort and relaxation. GI has been widely used in various medical diseases too. A systematic review of seven studies has successfully used GI to treat arthritis and other rheumatic disorders [22]. Another systematic review of 11 randomized controlled trials (RCTs) showed a significant reduction of non-musculoskeletal pain with GI [23].

Progressive Muscle Relaxation (PMR), well-known in the rehabilitative field, is based on tightening and relaxing specific muscle groups to achieve a positive state of relaxation [24, 25]. It has been studied to relieve symptoms, e.g., among women with breast cancer to reduce chemotherapy-induced nausea and vomiting [26, 27] and relieve pain associated with continuous passive motion in patients after total knee arthroplasty [28]. PMR could improve psychological well-being and quality of life in disorders including COVID-19 [29], HIV [30] and schizophrenia [31].

A study has reported beneficial effects of combined GI and PMR in chemotherapy patients [32]. However, no studies are currently available to determine the efficacy of GI and or PMR in treating bloating. Furthermore, no or limited studies are available in the assessment of mind-body physiological changes that are associated with the use of GI and or PMR. An investigation did show significant differences in skin temperature with PMR [33].

Thus, the current study is performed to address the following research questions:

1. What would be the script content of GI and PMR for bloating?

2. What are the physiological effects elicited on the brain event-related potentials (ERP) and heart rate variability (HRV) after listening to the audio scripts?

We postulated that GI is likely superior to PMR in achieving symptom control, improving psychological well-being, and quality of life for bloating. However, to test our hypothesis in a randomized trial, we will need to develop and validate the interventions first. In addition to psychometric validation, the current study also determines mind-body physiological responses as confirmatory validation.

## Methodology

To develop audio scripts, literature and established theoretical frameworks were reviewed first. A qualitative method using in-depth interviews was performed among participants with bloating to determine imagery themes. Subsequently, the GI and PMR audio scripts with background music were developed and validated using psychometric approaches. Lastly, for confirmatory validation, HRV and ERP measurements were determined in a group of healthy volunteers.

The study was approved by the Human Research Ethics Committee of Universiti Sains Malaysia (USM/JEPeM/20110562), and informed consent was obtained from all recruited participants.

### Development of audio scripts

**Review of literature and theoretical frameworks.**    For PMR, scripts from published studies were partly adapted to suit the local culture and language context [20, 24, 25, 34]. For the GI script, relevant studies were adopted to develop content for the in-depth interview [35, 36]. In addition, relevant theoretical frameworks were identified to base our GI intervention for bloating. These included the Perception Theory in Mind-Body Medicine [37], the Polyvagal theory [38], and the Biopsychosocial model [39]. The Perception Theory in Mind-Body Medicine suggests that mind-body interventions may play a role in realigning perceptual modalities by influencing the cognitive and beliefs system. Meanwhile, the Polyvagal Theory implies bio-mechanisms of phylogenetic vagal responses that affect gut physiological responses. Lastly, the biopsychosocial model reiterates a holistic, multimodal approach based on biology, psychology, and social intervention.

**In-depth interview.**    The in-depth interviews were conducted based on thematic analysis in the Malay language to derive culturally-appropriate bloating themes for the GI script [55, 56]. The interviews were transcribed verbatim and analyzed thematically by two researchers (VT and NA) using the NVivo software program (QSR International Pty Ltd., Melbourne, Australia). Invited participants fulfilled the Rome IV diagnostic criteria for functional bloating, were otherwise healthy, did not have any other chronic medical or psychiatric conditions, and were on stable medications for their bloating. The sample size was determined based on response saturation [40].

Two investigators checked and verified the transcripts and audio recordings for content and accuracy (VT, GK). After getting an overview of the content, further analysis of the texts was performed by generating codes, identifying categories, and clustering similar topics [41]. After that, the subcategories would be abstracted into generic and main categories. Codes were cross-checked, differences were discussed to resolve the disagreement, and if not determined, a third reviewer (YCK) would be consulted [42]. The findings from the thematic analysis were translated into English. Both language versions were cross-checked and verified by the investigators. Discrepancy and disagreement were resolved before finalizing the reported findings.

**Musical background and narration of the scripts.**    Using findings from in-depth interviews and literature, the GI and PMR scripts were then written (GK and VT). All the investigators discussed, revised, and agreed on the scripts. Subsequently, a local voice-over actor narrated the scripts in the local dialect. The voice, tempo, tone, clarity, and diction were carefully practiced to maximize engagement. Adequate pauses were inserted to allow time and space for listeners. Background music for the scripts was evaluated by seven experts (including one psychiatrist, one gastroenterologist, one health psychologist, and four musicians) from a list of 13 music of different genres. The evaluation was based on the Brunel Music Rating Inventory-2 (BMRI-2) [43], a 6-item questionnaire to assess the psychosocial effects of music, and the Affect Grid [44] to quantify the impact after listening to music. Subsequently, a professional composer (KS) composed a new background music from the chosen piece. All audios were recorded and edited using the Logic Pro X software (Apple Inc., USA).

### Validation of audio scripts

**Content- and face-validity of audio scripts.**    For content validity, experts from related fields were invited to rate the audios based on four domains (script content, narrations, experience, and clarity of instructions) using a 4-point Likert scale questionnaire. The rating questionnaire was adapted based on relevant literature [45–47]. For face validity, a structured interview session was conducted with healthy volunteers (mean age = 35.57 ± 14.28, 8 men, 24 women). Revisions were made to the audio scripts for subsequent pilot testing based on the responses and consensus from investigators.

**Physiological responses elicited from listening to audio scripts.** Healthy volunteers were recruited using purposive sampling. Exclusion criteria included acute and chronic medical and psychiatric illnesses, hearing impairment, and recent use of medications that may affect abdominal symptoms. Participants were invited to an air-conditioned, dimly lit room with only a maximum of three people to minimize noise. All electronic devices were turned off to reduce electrical wave interference. With the participant sitting upright on a chair, the MUSE EEG headband version 2016 (InterAxon Inc., Toronto, ON, Canada) (for ERP measurement) and Polar H10 HR sensor chest strap (Polar Electro UK Ltd., Warwick, UK) (for HRV measurement) were placed accordingly. Participants were first asked to listen to a 5-minute audio (control audio) that contained health information on abdominal bloating. Using a cross-over design, participants were randomized into either the GI or the PMR audio first, then cross-over to the other group in the same session after 15 min interval.

**ERP measurements.** The MUSE EEG headband has seven sensors with an adjustable band and is worn above the earlobes; three reference sensors (or Fpz) are positioned in the mid-forehead, one anterior frontal-7 (AF-7) sensor in the left forehead, one anterior frontal-8 (AF-8) in the right forehead, two above ear lobe, i.e., the left temporal-parietal-9 (TP-9) and the right temporal-parietal-10 (TP-10) [48–50]. These four sensors (AF-7 &8 and TP-9 & 10) would record the brain's alpha, beta, gamma, theta, and delta waves (**S1 File**). Recorded data would be transmitted to a Mind Monitor mobile app (Mind Monitor, USA) via BlueTooth. The absolute mean power for each frequency band was calculated by taking the average of four sensors, as shown in **Eq 1** below. Data were cleaned, and artifacts (ocular & muscular) were removed using Regression Methods before further analysis [65].

$$\text{Absolute Alpha power} = (\text{Alpha AF7} + \text{Alpha AF8} + \text{Alpha TP9} + \text{Alpha TP10}) \div 4 \quad \text{Eq1}$$

**HRV measurements.** The Polar H10 HR sensor, validated against the ECG gold standard [51], was positioned at the xiphoid process of the sternum using an adjustable chest strap and a Polar M400 monitoring wristband. The recorded R-R interval data was then transferred to the computer using the Polar FlowSync (version 2.5) for analysis. Before processing, the inter-beat intervals (IBIs) were manually corrected for ectopic or missed beats [52]. Kubios HRV software version 3.5 (Biosignal Analysis and Medical Imaging Group, Kuopio, Finland) was used to analyze the parameters. An Autoregressive (AR) algorithm was used for power spectral analysis of the frequency series.

**Data and statistical analysis.** All analyses were performed with the Statistical Package for the Social Science (SPSS) version 26 (IBM Corp., Armonk, NY, USA). Normally distributed values were reported in mean and standard deviation, while median and inter-quartile ranges were reported for non-normally distributed values. Pairwise comparison and repeated measure Analysis of Variance (RM-ANOVA) with Bonferroni adjustment were used for ERP variables. In contrast, Wilcoxon-signed rank tests and Kendall's W were used for HRV variables to determine the statistical differences between GI, PMR, and control audio. Significance was $P < 0.05$.

## Results

### Development of audio scripts

**In-depth interview.** Of 63 screened participants, 45 completed the interview that lasted half an hour (18 did not consent to the study). Three themes emerged, including 'balloon' being synonymous with the symptom of bloating, the inflated 'balloon' being the causation of bloating, and the inflated 'balloon' is associated with a sensation of pain or discomfort. All

**Table 1. Results of imagery description of bloating from the in-depth interview.**

| Domains | Themes | Quotations |
|---|---|---|
| Imagery description of bloating | Bloating as balloon | "I could literally feel my abdomen expand each time I breath. . .just like a balloon." (P28, 2020) |
| | | "My symptoms felt like a balloon in my stomach. It seems to be filled with lots of air, and I would burp to relieve it." (P10, 2020) |
| | | "I think the best description of the bloating symptom would be like having lots of air in the stomach, inflating like a balloon." (P17, 2020) |
| | | "I felt the bloatedness is like having air in the stomach. I usually felt it at the right side of my abdomen, just below the umbilicus. . ." (P07, 2020) |
| | An inflated balloon as the cause of bloating | "The bloating felt like a balloon inflating, always leading to pain and discomfort." (P16, 2020) |
| | | "The bloatedness felt like air in the stomach; sometimes it comes with abdominal distention and excessive farts." (P11, 2020) |
| | | "The bloating comes unpredictably. It always comes together with flatulence, burping, and fart. At times, I would feel there is this gas in my stomach, causing my abdomen to become distended." (P23, 2020) |
| | | "My bloating felt like having lots of air in my stomach, especially when it's empty (not eating anything), and I could also feel the abdomen getting bigger." (P29, 2020) |
| | Balloon sensation as pain or discomfort | "My bloating sensation comes with a lot of flatulence. I would also feel pain sometime,s and it would even lead to me having some difficulties in breathing." (P19, 2020) |
| | | "My bloating usually comes with a lot of symptoms. I would feel abdominal pain, breathlessness, and even migraine." (P24, 2020) |
| | | "I would feel as if there is some form of tightness in my chest, causing me to have difficulty in breathing, ng and sometimes it comes together with abdominal pain." (P03, 2020) |
| | | "It was like an expanding ball, on and it sounded like a drum when I percussed it (abdomen). Sometimes, it even made me felt breathlss, as if the lungs were being pushed upwards by the content." (P20, 2020) |

themes were verified based on the quotes. The themes and examples of quotes from participants for imagery description of bloating are shown in **Table 1**.

The results of the recent interview thus formed the basis of imagery intervention to reduce or deflate the 'balloon' in the script to improve bloating. Most of the participants described their bloating experiences as an episode of a "gaseous" event that is likened to "an expanding balloon," and some of them reported experiencing "difficulty in breathing" during their bloating episode. The abdomen would "eventually become bigger" and "sounded like a drum". Other aspects of the interview included precipitating factors, relieving factors, and impact, and the results were addressed in the script as part of the introduction phase to educate participants on bloating.

**Content of audio scripts.**    For the GI script, there were six parts, including the following: 1) introduction and instructions, 2) relaxation induction, 3) deepening relaxation and assurance, 4) imagery intervention, 5) gaining control and consolidation, and 6) awakening. For part 1, participants were instructed to accept "unpleasant sensations" that may arise and "be calm and confident" that their unpleasant feelings would "slowly fade away as they become more absorbed in their experience.". Part 2 would facilitate participants into the relaxation state with instructions including "take a deep breath. . .1,2,3,4,5 and breathe it out slowly. . .1,2,3,4,5. Let all the air in your lungs out". The third part involved instructions

including "for each gentle breath you take. . .you will drift deeper and deeper into relaxation. . ." and "you will find better health and greater control of your symptoms. . .". The fourth part would "imagine the balloon in your stomach expands.." and the fifth where participants would gradually gain control over their sensations, making it less intense and bothersome, e.g., "no matter how big the balloons get, you will feel that it does not bother you. . .and that feeling is going to fade away slowly. . .". In the final part, participants are slowly guided to the session's closure, "in the count of 5, you can slowly focus on your breath. . .whenever you are ready, you can open your eyes slowly. . .".

Similar to the GI script, the first part of PMR was the introduction and instructions, and the relaxation induction phase followed. The PMR intervention then commenced and involved progressive alternation between contraction and relaxation of different muscle groups, starting from the hands and torso, e.g., "clench your fist as tight as you can. . .", to the facial muscles, e.g., "close your eyes as tight as you can and hold it there for 5 seconds. . .", then neck and shoulder muscles, e.g. "now, focus on your shoulder. Lift your shoulder for 5 seconds. . .1,2,3,4,5 and relax. . .", and finally the abdominal muscle, e.g., "Now, focus on your shoulder. Feel your abdominal muscles and try to breathe in as much as you can. Hold it there for 5 seconds. . .1,2,3,4,5 and relax. . .". The final part was to guide participants to close the session.

**Narration and background music of audio scripts.** For the narration, the narrator's voice was normalized to -10dB and background noises were removed. The audio was camouflaged with binaural alpha waves pulse with f = 10 Hz. A frequency of 100 Hz was played on the left ear, while 110 Hz was played on the right ear at an acoustic pressure level = 73. For the background music, based on the BMRI-1, the most suitable was music no. 8 (Indonesian Instrumental) (average score 36), and likewise no. 8 was the best choice if based on the Affect Grid in the "happy" and "relaxed" grid (**Fig 1**). Hence, the traditional instrumental music genre was chosen as the theme. The final composed music consisted of traditional Malay instruments (*gambus* and *seruling*) in a major key melody and soundscapes from the natural forest, river flows, and birds. The tempo of the audio was set at 100–120 bpm, equivalent to the average heart rate, to rhythmically entrain and regulate the cardio-respiratory synchronisation while assisting relaxation [53, 54].

**Content and face-validity of audio scripts.** **Table 2** shows the results of content (CVI) and face validity (FVI) indexes for the four domains (scripts, narrations, audio experience, and clarity). Seven experts, including two staff nurses, two gastroenterology specialists, one health psychologist, and two experienced researchers, were invited for content validity. Overall, experts were generally satisfied, and there were several suggested revisions, including the following: 1) simplifying breathing during relaxation, 2) removing background noises and improving the narrator's voice, and 3) language. On point 1), the breathing instructions were modified to "breathe in and out calmly and slowly and take the time you need". On point 2), background noises were filtered using the Logic Pro X software and a deeper voice tone from the narrator to enhance specific keywords like "calm", "slowly", and "relax". For point 3), change to cultural appropriate terms and phrases, for instance, "perlahan" (slowly in standard language) with "*koh-koho*" (local dialect). CVIs for all domains ranged from 0.98 to 1.00 for the GI script and 0.96 to 1.00 for the PMR script (**Table 2**).

The face-validity from 32 participants with bloating (26 females; mean age 36.7 ± 14.9 years) were likewise positive for both scripts. For example, for the GI script, one participant described it as following: "it's like magic. . .I don't know how but it made me feel really good throughout the day". Some participants experienced slight discomfort but better after that, for example, a participant described the following: "I felt nauseating. . .after a while, the feeling wear off and I became much comfortable". There were shortcomings, including 1) unclear

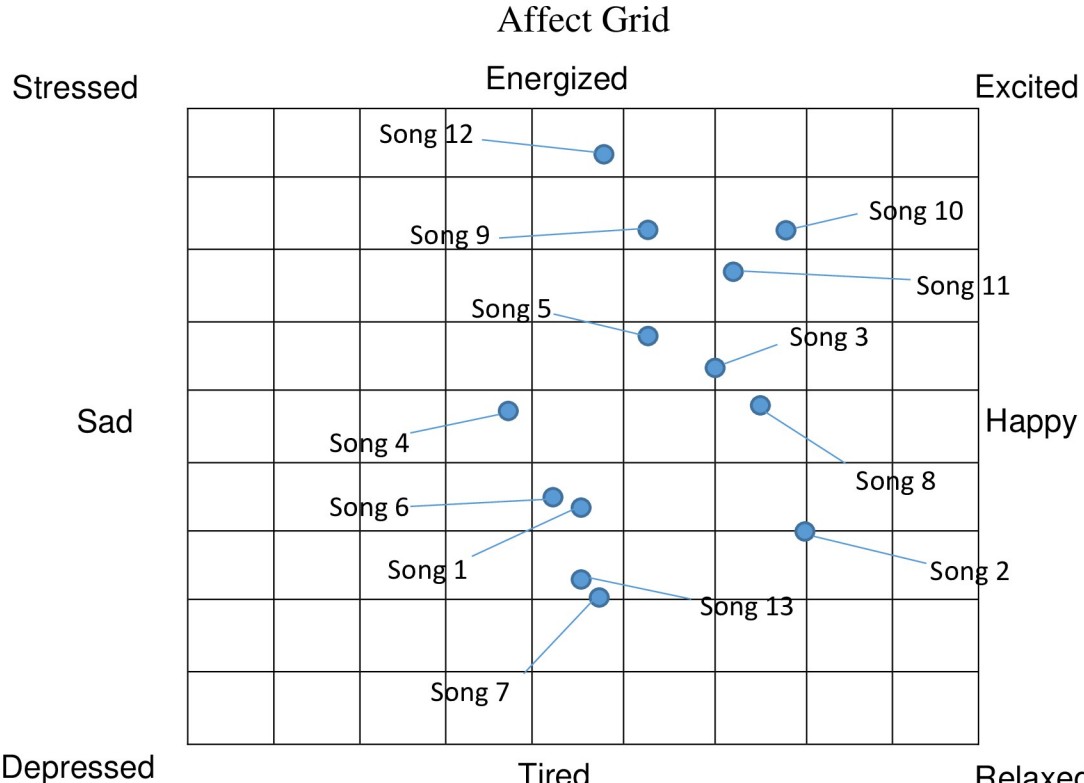

**Fig 1. The affect grid for the ratings of the 13 selected musical background.** Raters were instructed to rate how they are feeling right now and place one checkmark in the grid. The score is calculated according to the number of the squares. The horizontal dimension is used to score the pleasure-displeasure score, counting 1 to 9 starting at the left. The vertical dimension is used to score the arousal-sleepiness score, counting 1 to 9 starting at the bottom.

narration and background noises at times and 2) difficulties in the balloon imagery. The FVIs for the three domains (all except clarity) ranged from 0.95 to 0.98 for the GI script and 0.92 to 0.98 for the PMR script (**Table 2**).

**Confirmatory validation based on physiological responses.** Of 27 screened healthy volunteers, 20 participants were eventually recruited for the ERP and HRV measurements (7 did not consent to the study). Of the 20, 17 were analyzed (eight females, mean age of 23.4±1.7 years), and three were removed from the analysis as they fell asleep. Results are shown in **Fig 2** and **Table 3, respectively**.

Based on the pairwise comparison, ERP analysis revealed a significant mean difference between GI and PMR for alpha waves (p = 0.029), delta waves (p = 0.029), and between PMR and control for delta waves (p = 0.014) only (**Fig 2**). For HRV, using posthoc analysis, significant changes were seen between GI vs. control in the following parameters: RR (p = 0.049), LF (p = 0.017), TP (p = 0.044), and LFHF (p = 0.048), and between PMR vs control in the following parameters: RR (p = 0.031), LF (p = 0.002), TP (p = 0.007) and LFHF (p = 0.002) (**Table3**).

## Discussion

The following is a summary of the main findings: 1) bloating, described as an inflated 'balloon' during the in-depth interview, forms the basis of imagery script for intervention, i.e., deflating the 'balloon', 2) newly developed GI and PMR audio scripts with background music demonstrated appropriate content and face validity, 3) there are differences in the ERP and HRV

**Table 2. Content and face validity of audio scripts.**

| Section/ Type of instruments | I-CVI | | I-FVI | |
|---|---|---|---|---|
| | GI | PMR | GI | PMR |
| **Domain 1: Script content** | | | | |
| Helps to change attention focus to decrease symptom experience | 0.86 | 1.00 | 0.97 | 0.88 |
| Alters the perceptual experience of the symptoms | 1.00 | 1.00 | 0.91 | 0.84 |
| Helps in providing suggestions for overall increased sense of health & comfort | 1.00 | 1.00 | 0.97 | 0.97 |
| Helps in providing suggestions for the intestines to become immune to irritation or upsetting life events | 1.00 | 0.86 | 0.84 | 0.84 |
| Helps in providing suggestions and imagery to encourage normal & healthy bowel functioning | 1.00 | 0.86 | 1.00 | 0.97 |
| Instructions are clear & comprehensible | 1.00 | 1.00 | 0.97 | 0.97 |
| Sequencing & length is appropriate | 1.00 | 1.00 | 0.97 | 1.00 |
| Average | 0.98 | 0.96 | 0.95 | 0.92 |
| **Domain 2: Narrations** | | | | |
| The tone of the narrator is appropriate | 1.00 | 1.00 | 1.00 | 1.00 |
| The fluency & clarity of the narration are appropriate | 1.00 | 1.00 | 1.00 | 1.00 |
| The pace & speed of the narration are appropriate | 1.00 | 1.00 | 0.91 | 0.97 |
| The language/dialect used is appropriate | 1.00 | 1.00 | 1.00 | 0.94 |
| The recording quality of the narrations is appropriate | 1.00 | 1.00 | 1.00 | 1.00 |
| Average | 1.00 | 1.00 | 0.98 | 0.98 |
| **Domain 3: Audio experience** | | | | |
| I was able to finish the hearing session without falling asleep | 1.00 | 1.00 | 0.97 | 0.91 |
| I was able to follow through the hearing session | 1.00 | 1.00 | 0.97 | 0.97 |
| I was able to imagine well / feel the relax sensation after tensing my muscles | 1.00 | 1.00 | 0.94 | 0.97 |
| I felt affected by the imagery / progressive muscle-induced relaxation | 1.00 | 1.00 | 0.94 | 0.97 |
| I will recommend this to my friends & family | 1.00 | 1.00 | 0.94 | 0.91 |
| Average | 1.00 | 1.00 | 0.95 | 0.95 |
| **Domain 4: Script clarity** | | | | |
| Introduction | 1.00 | 1.00 | - | - |
| Relaxation breathing | 1.00 | 1.00 | - | - |
| Deepening (GI)—Upper Limbs Relaxation (PMR) | 1.00 | 1.00 | - | - |
| Imaginary Part 1 (GI)—Facial Relaxation (PMR) | 0.86 | 1.00 | - | - |
| Suggestions (GI)—Neck Relaxation (PMR) | 1.00 | 1.00 | - | - |
| Awakening | 1.00 | 1.00 | - | - |
| Average | 0.98 | 1.00 | - | - |

Note: GI: Guided-imagery, PMR: Progressive muscle relaxation, I-CVI: Item-level content validity index, I-FVI: Item-level face validity index.

physiological responses between the two interventions, providing confirmatory validation. Overall, the alpha and delta waves differ between GI and PMR, but GI and PMR have similar autonomic responses.

Development of imagery script involves an initial literature review, in-depth interview, and stepwise modifications by investigators based on comments from various experts. Bloating as a concept can be interpreted differently across different languages and cultures. The sensation of an inflated 'balloon', synonymous with bloating, is cross-cultural. The metaphor allows GI intervention to 'deflate' the balloon, which will relieve bloating. This aligns with a popular Ericksonian psychotherapy strategy where scientific-based therapy was integrated into stories, anecdotes, information, and metaphors [55]. Certain words, including "calm" and "slowly,"

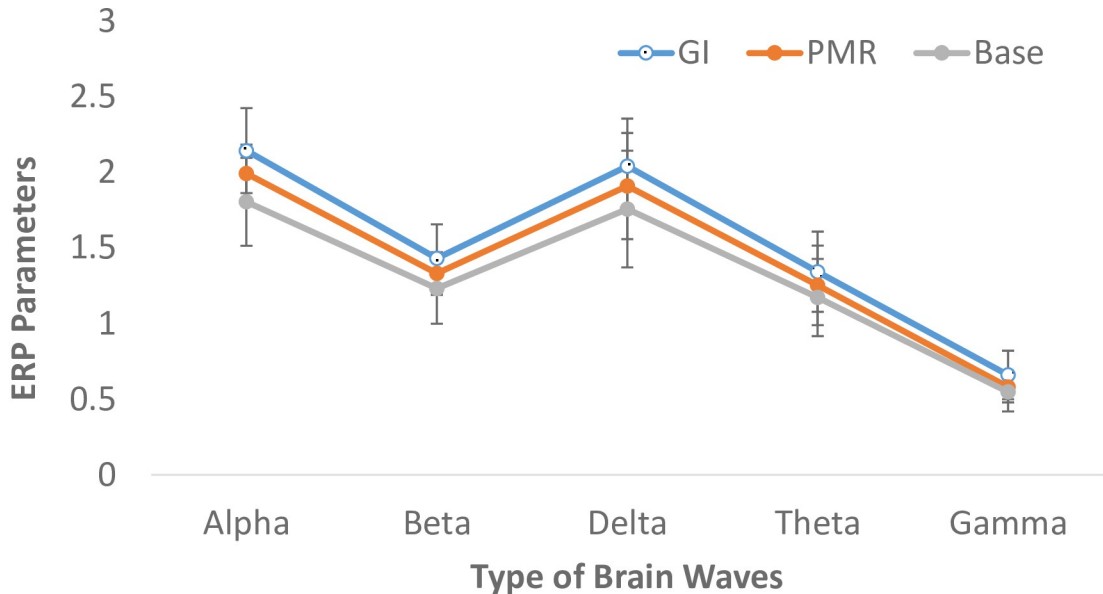

| Audio intervention | GI. Mean (SD.) | PMR Mean (SD.) | Control (Base) Mean (SD.) |
|---|---|---|---|
| | ERP parameters | | |
| Alpha | 2.14 (0.28)* | 1.99 (0.19)* | 1.80 (0.29) |
| Beta | 1.43 (0.22) | 1.33 (0.14) | 1.23 (0.23) |
| Delta | 2.04 (0.31)* | 1.90 (0.35)* | 1.75 (0.38)* |
| Theta | 1.34 (0.26) | 1.25 (0.26) | 1.17 (0.25) |

**Fig 2. Reported values, pairwise comparison, and RM-ANOVA analysis of ERP parameters between guided imagery, progressive muscle relaxation and controls.** GI: Guided imagery, PMR: Progressive muscle relaxation, ERP: Brain event-related potentials.
*Significant mean difference based on pairwise comparison in alpha waves between GI and PMR ($p = 0.029$), delta waves between GI and PMR ($p = 0.029$) and between PMR and control ($p = 0.014$).

are repetitively mentioned in both scripts to provide soothing, relaxing effects and to enhance beneficial outcomes from increased exposure to unpleasant 'balloon' sensations [56].

The development and validation of scripts were based on a robust methodology as per recommendations of the Medical Research Council [57]. A robust and validated script would enhance reliability, acceptability, feasibility, and compliance, allowing the retention of research subjects and consistent intervention delivery. In addition to scripts, the background music was incorporated, and music is a common addition and/or standalone tool to many psychological interventions to enhance therapeutic outcomes and to facilitate delivery [17, 24, 58–62]. At the forefront, we have considered the use of language and culturally appropriate background music, in alignment with the audio interventions [63–67]. Our present study employed a validated method to select a suitable background music. The process included the use of BMRI-2 scale and the Affect Grid, which allow rating of "emotions" that offer quantitative and qualitative scores, respectively. The complete narrated script with music as a product was then tested for content and face validity. The calculated indexes (CVI and FVI) indicated a robust product.

The products were also tested for effects on brainwaves and autonomic responses as confirmatory validation since a robust intervention should produce appropriate physiological responses. Each brain wave pattern may be affected differently by certain conditions, for e.g.,

**Table 3. Comparison of heart rate variables between guided imagery, progressive muscle relaxation, and control audio.**

| Variables | Group | Median (IQR) | Comparison | Z-statistics | Effect size[a] | p-value[b] |
|---|---|---|---|---|---|---|
| RR | GI | 797.0 (100.0) | GI VS PMR | -0.92 | 0.170 | 0.356 |
|  | PMR | 774.0 (158.5) | GI VS C | -1.97 |  | 0.049* |
|  | C | 780.0 (138.5) | PMR VS C | -2.15 |  | 0.031* |
| HR | GI | 75.0 (10) | GI VS PMR | -1.69 | 0.145 | 0.091 |
|  | PMR | 78.0 (14.5) | GI VS C | -1.86 |  | 0.062 |
|  | C | 77.0 (13.5) | PMR VS C | -1.38 |  | 0.168 |
| PNSI | GI | -0.39 (1.2) | GI VS PMR | -0.36 | 0.014 | 0.723 |
|  | PMR | -0.70 (1.3) | GI VS C | -1.30 |  | 0.193 |
|  | C | -0.19 (1.6) | PMR VS C | -0.71 |  | 0.478 |
| SNSI | GI | 0.3 (1.4) | GI VS PMR | -0.21 | 0.024 | 0.831 |
|  | PMR | 0.5 (1.4) | GI VS C | -0.31 |  | 0.758 |
|  | C | 0.5 (1.3) | PMR VS C | -0.99 |  | 0.320 |
| LF | GI | 1144.0 (1514.0) | GI VS PMR | -0.45 | 0.170 | 0.653 |
|  | PMR | 1109.0(1023.5) | GI VS C | -2.39 |  | 0.017* |
|  | C | 766.0 (433.0) | PMR VS C | -3.10 |  | 0.002* |
| HF | GI | 447.0 (508.5) | GI VS PMR | -0.45 | 0.014 | 0.653 |
|  | PMR | 487.0 (837.5) | GI VS C | -0.64 |  | 0.523 |
|  | C | 443.0 (447.5) | PMR VS C | -1.63 |  | 0.102 |
| TP | GI | 1648.0 (2301.0) | GI VS PMR | -0.26 | 0.149 | 0.795 |
|  | PMR | 1888.0 (1746.5) | GI VS C | -2.01 |  | 0.044* |
|  | C | 1255.0 (546.0) | PMR VS C | -2.68 |  | 0.007* |
| LF/HF ratio | GI | 2.7 (2.4) | GI VS PMR | -1.40 | 0.180 | 0.163 |
|  | PMR | 3.3 (2.8) | GI VS C | -1.68 |  | 0.048* |
|  | C | 2.1 (2.3) | PMR VS C | -3.10 |  | 0.002* |

GI: Guided imagery, PMR: Progressive muscle relaxation, C: Control audio, HRV: Heart rate variability, HF: High frequency, LF: Low frequency, MeanHR: Heart rate, MeanRR: Respiratory rate, PNSI: Parasympathethic nervous system index, SNSI: Sympathetic nervous sytem indicator, TP: Total power.

[a]Kendall's W

[b]Wilcoxon-signed rank test

*significant mean differences.

meditation enhances theta waves and hypnosis induces alpha waves [68]. It was found in the current study that alpha and delta waves were significantly different between GI vs PMR. Imagery is likened to hypnosis but without affecting consciousness. Thus, it may not be surprising that the alpha waves were different between GI and PMR. Inducing alpha waves could also reduce stress [69] and help depression [54]. Delta wave is usually seen during deep sleep, coma, or anaesthesia [70] and represents comfort and reduced pain [71]. The significant difference with delta waves in GI vs PMR, and PMR vs control, indicated that GI and PMR are interventions that could reduce painful sensations like the 'balloon'. In addition, recent studies have correlated delta waves with cognitive process such as attention, problem-solving, and perception [72]. For intervention, especially GI and, to a lesser extent PMR, involves significant cognitive function to execute a deflation of a balloon or relax the muscles. Background music and bianural beats might also explain the effects seen on brainwaves [73–78]. Various research has shown that music was able to increase alpha and decrease beta waves among individuals with depression [54, 77] or anxiety [18, 54].

HRV analysis also confirmed that GI and PMR were different from controls. First, the RRs were significantly different from controls, since GI and PMR required adherence to breathing

instructions. Second, low-frequency (LF) power (0.04–0.15 Hz), measuring lower paced respiratory activities on autonomic responses, the LF/HF ratio measuring the sympatho-vagal balance [79, 80], and total power measuring overall respiratory responses on autonomic activities were also different from controls. It can be concluded that both GI and PMR induce similar physiological responses to autonomic activities through regulated breathing patterns, which are also an indirect indicator of a relaxation state. For example, Edmond et al. found that an increased LF range correlated with ease and higher comfort among participants with slow-paced breathing [81]. Similarly, Lin et al. reported an increase in LF power and LF/HF ratio correlated with an increased perception of relaxation in the paced breathing group [82].

Our study has a few limitations. First, we recognize that different individual has different interpretation and execution of mind-body techniques. This factor may impact therapy outcomes, especially GI, and future studies should incorporate a validated tool to select participants [83, 84]. Second, this study only involved a limited number of participants because of its experimental and exploratory nature. Third, HRV responses might be confounded by other factors besides breathing, including sleep quality which was not covered in our study. Furthermore, HRV responses were conducted while participants were in a 'relaxed' state and not while they were 'stressed' or experiencing bloating, which might explain the absence of changes in HR, SNSI and PNSI parameters. Fourth, our interviews were conducted using the Malay language, translated into English, and cross-checked by experts but not participants. Finally, we acknowledge that music is cultural bound, and thus the current choice of background music may not be suitable across different population. Hence, it would be prudent to conduct similar studies among people from different countries and cultures.

In conclusion, based on qualitative interviews, bloating is found synonymous with inflated 'balloon' and the metaphor may provide a suitable imagery intervention. In addition, the newly developed GI and PMR audio scripts with background music have been validated using psychometric and physiologic (ERP and HRV) responses. Future studies would include a randomized trial to compare the two techniques in treating bloating.

## Supporting information

**S1 File. Supporting information–contains all the supporting tables and figures.**
(DOCX)

## Acknowledgments

Part of the work herein have been presented at the Asian Pacific Digestive Week (APDW) 2021, 19–22 August 2021 and published as conference-related abstract: Development of cultural specific guided imagery and progressive muscle relaxation therapy for treatment of functional bloating. Journal of Gastroenterology and Hepatology 36 (Suppl. 2), pg156. https://doi.org/10.1111/jgh.15607. The work has also been presented at the YSN-ASM International Scientific Virtual Conference (ISVC) 2021, 29 March– 1 April and published as conference-related abstract: Development of Guided Imagery and Progressive Muscle Relaxation Therapy Audio for Patients with Functional Bloating. ASM Science Journal 16, pg13. https://doi.org/10.32802/asmscj.2021.isvc. Also, we wish to thank the staffs of GI Function & Motility Unit of Hospital USM, and others for their assistance and input throughout the study.

## Author Contributions

**Conceptualization:** Vincent Tee, Garry Kuan, Yee Cheng Kueh, Nurzulaikha Abdullah, Kamal Sabran, Nashrulhaq Tagiling, Nur-Fazimah Sahran, Yeong Yeh Lee.

**Data curation:** Vincent Tee, Garry Kuan, Yee Cheng Kueh, Nurzulaikha Abdullah, Nur-Fazimah Sahran, Yeong Yeh Lee.

**Formal analysis:** Vincent Tee, Yee Cheng Kueh, Nurzulaikha Abdullah, Nur-Fazimah Sahran.

**Funding acquisition:** Vincent Tee, Garry Kuan, Yee Cheng Kueh, Yeong Yeh Lee.

**Investigation:** Vincent Tee, Garry Kuan, Yee Cheng Kueh, Nurzulaikha Abdullah, Kamal Sabran, Nur-Fazimah Sahran, Yeong Yeh Lee.

**Methodology:** Vincent Tee, Garry Kuan, Yee Cheng Kueh, Nurzulaikha Abdullah, Nur-Fazimah Sahran, Yeong Yeh Lee.

**Project administration:** Vincent Tee, Garry Kuan, Yee Cheng Kueh, Nurzulaikha Abdullah, Nur-Fazimah Sahran, Tengku Ahmad Iskandar Tengku Alang, Yeong Yeh Lee.

**Resources:** Vincent Tee, Garry Kuan, Yee Cheng Kueh, Kamal Sabran, Tengku Ahmad Iskandar Tengku Alang, Yeong Yeh Lee.

**Software:** Kamal Sabran, Tengku Ahmad Iskandar Tengku Alang.

**Supervision:** Garry Kuan, Yee Cheng Kueh, Yeong Yeh Lee.

**Validation:** Garry Kuan, Yee Cheng Kueh, Kamal Sabran, Tengku Ahmad Iskandar Tengku Alang, Yeong Yeh Lee.

**Visualization:** Vincent Tee, Garry Kuan, Yee Cheng Kueh, Kamal Sabran, Yeong Yeh Lee.

**Writing – original draft:** Vincent Tee, Garry Kuan, Yee Cheng Kueh, Nashrulhaq Tagiling, Yeong Yeh Lee.

**Writing – review & editing:** Vincent Tee, Garry Kuan, Yee Cheng Kueh, Nashrulhaq Tagiling, Yeong Yeh Lee.

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
