## [Decision Letter · Decision Letter 0]

25 May 2022

PONE-D-22-12725Development and Validation of Audio-based Guided Imagery and Progressive Muscle Relaxation Tools for Functional BloatingPLOS ONE

Dear Dr. Lee,

Thank you for submitting your manuscript to PLOS ONE. After careful consideration, we feel that it has merit but does not fully meet PLOS ONE’s publication criteria as it currently stands. Therefore, we invite you to submit a revised version of the manuscript that addresses the points raised during the review process.

ACADEMIC EDITOR: 

First of all, I would like to thank you for submitting this well-structured study to our journal that addresses a health issue that I believe has a significant negative impact on overall quality of life.

I would like to inform you that I fully agree with the constructive criticism and guiding comments made by the reviewers with a careful reviewing your manuscript and that it requires some revisions in this context.

I look forward to receive your revised manuscript, which you will prepare in line with the reviewers' suggestions, for reconsideration before publication.

We look forward to receiving your revised manuscript.

Kind regards,

Sukru Torun

Academic Editor

PLOS ONE

Journal Requirements:

3. We noted in your submission details that a portion of your manuscript may have been presented or published elsewhere. 

"Part of the work herein have been presented at the Asian Pacific Digestive Week (APDW) 2021, 19-22 August 2021 and published as conference-related abstract: Development of cultural specific guided imagery and progressive muscle relaxation therapy for treatment of functional bloating. Journal of Gastroenterology and Hepatology 36 (Suppl. 2), pg156. https://doi:10.1111 /jgh.15607. The work has also been presented at the YSN-ASM International Scientific Virtual Conference (ISVC) 2021, 29 March – 1 April and published as conference-related abstract: Development of Guided Imagery and Progressive Muscle Relaxation Therapy Audio for Patients with Functional Bloating. ASM Science Journal 16, pg13. " ext-link-type="uri" xlink:type="simple">https://doi.org/10.32802/asmscj.2021.isvc."

6. Please upload a copy of Figure 2, to which you refer in your text on page 10. If the figure is no longer to be included as part of the submission please remove all reference to it within the text.

Reviewers' comments:

Reviewer's Responses to Questions

**Comments to the Author**

1. Is the manuscript technically sound, and do the data support the conclusions?

Reviewer #1: Yes

Reviewer #2: Yes

2. Has the statistical analysis been performed appropriately and rigorously? 

Reviewer #1: Yes

Reviewer #2: Yes

3. Have the authors made all data underlying the findings in their manuscript fully available?

Reviewer #1: Yes

Reviewer #2: Yes

4. Is the manuscript presented in an intelligible fashion and written in standard English?

Reviewer #1: Yes

Reviewer #2: Yes

5. Review Comments to the Author

Reviewer #1: Thank you for submitting this paper for publication consideration. The topic is important the authors put a lot of effort into this project.

Introduction

• First sentence reads awkwardly – please rephrase

• Suggest including specific research questions (i.e., research questions 1, 2, 3) to help the readers understand the various tasks the authors were trying to accomplish. These research questions could then be used to structure the Method and Results sections.

Method

• What type of qualitative data analytic technique did the authors use (content analysis, thematic analysis, grounded theory, etc.)? Why did they use this technique? The rationale is important.

• Did participants member check transcripts? Did participants provide trustworthiness? If not, these are limitations of the study. In future research, perhaps people with the lived experience of bloating could be more involved in other aspects of the research as they are experts.

• Were interviews conducted in English? Were they translated/interpreted?

• Regarding content validity, why was a service user or person with bloating not included as an expert? Perhaps include this as a limitation.

Results

• Suggest including effect sizes as well as p-values in Table 3.

Discussion

• What are implications for clinical practice?

• What are suggestions for future research?

• A potential limitation is the music itself as music is culturally bound. The music may not work with people from other cultures. I believe this is an important limitation that the author should consider integrating into their limitations section.

Reviewer #2: Thank you for the opportunity to review this manuscript. The authors have done very good work designing and executing a well-constructed study to address a health concern that has a significant impact on quality of life and other dimensions of health. The authors represent an experienced and multidisciplinary team well prepared to conduct this type of study.

Overall, the authors have done excellent work in describing their study with great detail and clarity. They have been clear describing each step of the study, their process and efforts to develop the tools and resources to address the issue of bloating and the measures they implemented in the study.

On page 2 (last paragraph) the authors have cited no literature related to guided imagery. The literature cited in this paragraph is only related to progressive muscle relaxation. There is a good body of evidence surrounding the use of guided imagery and this should be integrated into the manuscript. Given the study is focused on development and validation of audio-based guided imagery, literature related to guided imagery needs to be reviewed to support the development of the guided imagery intervention. The authors indicate the scripts were developed based on literature reviews. Sources should be cited in the introduction.

On page 3, in the review of literature and theoretical frameworks section there is limited review of literature, rather it focuses only on studies that were adopted to develop content. There is no review of literature that indicates a review of existing literature that further helped inform the researchers conceptualization of this study. This review of literature is so brief, I would recommend providing more context… this may need to go in the introduction rather than in the method section.

The authors have done very good work indicating the importance of cultural considerations (language and music) and how they integrated current research with these considerations at the forefront. It is important to also draw upon literature surrounding this to cite this in the manuscript.

The authors have provided a clear description of the study methods and each of the measurement tools. The authors do not detail how the imagery themes were determined. Detail is needed for readers to understand how the authors analyzed this qualitative data from study participants.

On page 12, Line 316, the authors state that the development of imagery scripts involves a thorough literature review. This is not evident from what is represented in this manuscript. While the body of literature surrounding the use of GI may be limited for treating bloating, the authors should search related literature to support the use of GI in their study.

Addressing the issues surrounding the literature as it relates to the topic will help to strength the manuscript and support the strong claims of having conducted a thorough review of the literature.

6. PLOS authors have the option to publish the peer review history of their article (what does this mean?). If published, this will include your full peer review and any attached files.

Reviewer #1: No

Reviewer #2: **Yes: **Annie Heiderscheit, Ph.D., MT-BC, LMFT

---

## [Author Response · Author response to Decision Letter 0]

8 Jul 2022

Reviewer 1: 

1. First sentence reads awkwardly – please rephrase

- Thank you for the suggestions. We have rephrased and made changes to the first sentence.

2. Suggest including specific research questions (i.e., research questions 1, 2, 3) to help the readers understand the various tasks the authors were trying to accomplish. These research questions could then be used to structure the Method and Results sections.

- Specific research questions have been added at pg.3, line 80-82 as per suggested. The Methods and Results section were restructured as well. 

3. What type of qualitative data analytic technique did the authors use (content analysis, thematic analysis, grounded theory, etc.)? Why did they use this technique? The rationale is important.

- Thank you very much for the comments in the method sections. We have included the type of qualitative data analysis (thematic analysis) and its rationale (pg.4, line 114-115).

4. Did participants member check transcripts? Did participants provide trustworthiness? If not, these are limitations of the study. In future research, perhaps people with the lived experience of bloating could be more involved in other aspects of the research as they are experts.

- Participants did not check the transcripts nor provide trustworthiness. These limitations were added (pg.14, line 393-394).

Yes, we do agree with the suggestion on involving people with the lived experience of bloating in future research, this has been added in the discussions section (pg. 14, line 402-403). 

5. Were interviews conducted in English? Were they translated/interpreted?

- Interviews were conducted in the Malay Language. Transcripts were translated into English by the transcribers prior to the analysis. (pg.4, line 114-115 and line 128).

6. Regarding content validity, why was a service user or person with bloating not included as an expert? Perhaps include this as a limitation.

- Thank you for the comment. The term “expert” and “content” validity in psychometric studies are not interpreted literally. “Experts” in content validation are limited to researchers/clinicians that are qualified or experienced in related fields. Service user/person with bloating/patients/participants should be considered in the face/construct/response validation. 

Further explanations on content validation and face validation can be found in the references below. Also, we have already included service participants with bloating in the audio script generation prior to content (expert) validation as well as the psychometric testing for the physiological changes (brainwave and heart rate). 

Yusoff MSB. ABC of content validation and content validity index calculation. 

Education in Medicine Journal. 2019;11(2):49–54. https://doi.org/10.21315/eimj2019.11.2.6

Yusoff MSB. ABC of response process validation and face validity index 

calculation. Education in Medicine Journal. 2019;11(3):55–61. https://doi.org/10.21315/

eimj2019.11.3. 

7. Suggest including effect sizes as well as p-values in Table 3.

- We have included the effect sizes in Table 3 (pg.11-12).

8. What are implications for clinical practice?

What are suggestions for future research?

A potential limitation is the music itself as music is culturally bound. The music may not work with people from other cultures. I believe this is an important limitation that the author should consider integrating into their limitations section.

- Thank you for your suggestions. Bloating is found synonymous with inflated ‘balloon’ and the clinical implication of this metaphor is that it may provide a suitable imagery intervention. In addition, the newly developed GI and PMR audio scripts with background music have been validated using psychometric and physiologic (ERP and HRV) responses, and are available for future research. Future studies would include a randomized trial to compare the two techniques in treating bloating. We have included the above alongside conclusion section of the manuscript. 

We acknowledged the potential limitation of cultural differences of music and include into the limitations (page 14, line 394-397).

Reviewer 2: 

1. On page 2 (last paragraph) the authors have cited no literature related to guided imagery. The literature cited in this paragraph is only related to progressive muscle relaxation. There is a good body of evidence surrounding the use of guided imagery and this should be integrated into the manuscript. Given the study is focused on development and validation of audio-based guided imagery, literature related to guided imagery needs to be reviewed to support the development of the guided imagery intervention. The authors indicate the scripts were developed based on literature reviews. Sources should be cited in the introduction.

On page 3, in the review of literature and theoretical frameworks section there is limited review of literature, rather it focuses only on studies that were adopted to develop content. There is no review of literature that indicates a review of existing literature that further helped inform the researchers conceptualization of this study. This review of literature is so brief, I would recommend providing more context… this may need to go in the introduction rather than in the method section.

- Thank you for the suggestion. We have added relevant literature reviews at the introduction section (pg.2-3, line 57-78). 

2. The authors have done very good work indicating the importance of cultural considerations (language and music) and how they integrated current research with these considerations at the forefront. It is important to also draw upon literature surrounding this to cite this in the manuscript.

- Thank you very much for the positive comment. We have included relevant literatures (pg. 13, line 343-349)

3. The authors have provided a clear description of the study methods and each of the measurement tools. The authors do not detail how the imagery themes were determined. Detail is needed for readers to understand how the authors analyzed this qualitative data from study participants.

- We have added explanations on the methodology sections that delineate how we analyzed the qualitative data and eventually came up with the imagery theme (pg.7, line 216-223).

4. On page 12, Line 316, the authors state that the development of imagery scripts involves a thorough literature review. This is not evident from what is represented in this manuscript. While the body of literature surrounding the use of GI may be limited for treating bloating, the authors should search related literature to support the use of GI in their study.

- Thank you for the feedback. We have included literatures that utilised/integrated GI in treating patients with various medical diseases in the introduction section (pg.2-3, line 57-65).

---

## [Decision Letter · Decision Letter 1]

11 Aug 2022

Development and Validation of Audio-based Guided Imagery and Progressive Muscle Relaxation Tools for Functional Bloating

PONE-D-22-12725R1

Dear Dr. Lee,

We’re pleased to inform you that your manuscript has been judged scientifically suitable for publication and will be formally accepted for publication once it meets all outstanding technical requirements.

Kind regards,

Jianhong Zhou

Staff Editor

PLOS ONE

Additional Editor Comments (optional):

Reviewers' comments:

Reviewer's Responses to Questions

**Comments to the Author**

1. If the authors have adequately addressed your comments raised in a previous round of review and you feel that this manuscript is now acceptable for publication, you may indicate that here to bypass the “Comments to the Author” section, enter your conflict of interest statement in the “Confidential to Editor” section, and submit your "Accept" recommendation.

Reviewer #1: All comments have been addressed

2. Is the manuscript technically sound, and do the data support the conclusions?

Reviewer #1: Yes

3. Has the statistical analysis been performed appropriately and rigorously? 

Reviewer #1: Yes

4. Have the authors made all data underlying the findings in their manuscript fully available?

Reviewer #1: Yes

5. Is the manuscript presented in an intelligible fashion and written in standard English?

Reviewer #1: Yes

6. Review Comments to the Author

Reviewer #1: Thank you for your openness to feedback. I wish the authors the best and hope to read more from them in the future.

7. PLOS authors have the option to publish the peer review history of their article (what does this mean?). If published, this will include your full peer review and any attached files.

Reviewer #1: No

---

## [Editor Report · Acceptance letter]

15 Aug 2022

PONE-D-22-12725R1 

Development and Validation of Audio-based Guided Imagery and Progressive Muscle Relaxation Tools for Functional Bloating 

Dear Dr. Lee:

I'm pleased to inform you that your manuscript has been deemed suitable for publication in PLOS ONE. Congratulations! Your manuscript is now with our production department. 

Kind regards, 

on behalf of

Jianhong Zhou 

Staff Editor

PLOS ONE